# Biallelic Variants in *KIF17* Associated with Microphthalmia and Coloboma Spectrum

**DOI:** 10.3390/ijms22094471

**Published:** 2021-04-25

**Authors:** Antonella Riva, Antonella Gambadauro, Valeria Dipasquale, Celeste Casto, Maria Domenica Ceravolo, Andrea Accogli, Marcello Scala, Giorgia Ceravolo, Michele Iacomino, Federico Zara, Pasquale Striano, Caterina Cuppari, Gabriella Di Rosa, Maria Concetta Cutrupi, Vincenzo Salpietro, Roberto Chimenz

**Affiliations:** 1Unit of Medical Genetics, IRCCS Giannina Gaslini Institute, Via Gerolamo Gaslini 5, 16147 Genoa, Italy; riva.anto94@gmail.com (A.R.); scarsoacco@hotmail.com (A.A.); micheleiacomino@gaslini.org (M.I.); Federico.zara@unige.it (F.Z.); 2Department of Neurosciences Rehabilitation, Ophthalmology, Genetics, Maternal and Child Health (DiNOGMI), University of Genoa, Via Gerolamo Gaslini 5, 16147 Genoa, Italy; marcelloscala87@gmail.com (M.S.); strianop@gmail.com (P.S.); 3Department of Human Pathology in Adult and Developmental Age “Gaetano Barresi”, Unit of Emergency Pediatric, University of Messina, Via Consolare Valeria 1, 98124 Messina, Italy; gambadauroa92@gmail.com (A.G.); dipasquale.valeria@libero.it (V.D.); celestecasto@libero.it (C.C.); maria.domenica.ceravolo@gmail.com (M.D.C.); giorgiaceravolo@gmail.com (G.C.); katia.cuppari@libero.it (C.C.); cutrupimaricia@gmail.com (M.C.C.); 4Department of Paediatrics, Division of Medical Genetics, McGill University, Montreal, QC H3A 2R7, Canada; 5Pediatric Neurology and Muscular Diseases Unit, IRCCS Giannina Gaslini Institute, Via Gerolamo Gaslini 5, 16147 Genoa, Italy; 6Department of Human Pathology in Adult and Developmental Age “Gaetano Barresi”, Division of Child Neurology and Psychiatry, University of Messina, Via Consolare Valeria 1, 98124 Messina, Italy; gdirosa@unime.it; 7Department of Human Pathology in Adult and Developmental Age “Gaetano Barresi, Unit of Pediatric Nephrology and Dialysis, University of Messina, Via Consolare Valeria 1, 98124 Messina, Italy; roberto.chimenz@unime.it

**Keywords:** *KIF17*, microphthalmia, coloboma, MAC spectrum, congenital eye defects

## Abstract

Microphthalmia, anophthalmia, and coloboma (MAC) are a group of congenital eye anomalies that can affect one or both eyes. Patients can present one or a combination of these ocular abnormalities in the so called “MAC spectrum”. The *KIF17* gene encodes the kinesin-like protein Kif17, a microtubule-based, ATP-dependent, motor protein that is pivotal for outer segment development and disc morphogenesis in different animal models, including mice and zebrafish. In this report, we describe a Sicilian family with two siblings affected with congenital coloboma, microphthalmia, and a mild delay of motor developmental milestones. Genomic DNA from the siblings and their unaffected parents was sequenced with a clinical exome that revealed compound heterozygous variants in the *KIF17* gene (NM_020816.4: c.1255C > T (p.Arg419Trp); c.2554C > T (p.Arg852Cys)) segregating with the MAC spectrum phenotype of the two affected siblings. Variants were inherited from the healthy mother and father, are present at a very low-frequency in genomic population databases, and are predicted to be deleterious in silico. Our report indicates the potential co-segregation of these biallelic *KIF17* variants with microphthalmia and coloboma, highlighting a potential conserved role of this gene in eye development across different species.

## 1. Introduction

The *KIF17* gene (OMIM *605037) is located at chromosome 1p36.12 and encodes the kinesin-like protein Kif17, a microtubule-based, ATP-dependent motor protein [1,2]. The homodimeric Kif17 is the mammalian homolog of the molecular motor OSM-3, a dendritic motor for odorant receptors in *Caenorhabditis elegans* and holds many different biological functions ranging from the in-vesicles dendritic transport of N-methyl-D-aspartate receptor subunit 2B (NR2B), kainate receptor subunit GluR5, and voltage-gated K+ channel Kv4.2, to intraflagellar transport (IFT)/ciliogenesis and spermatogenesis [3]. Eventually, in vertebrate photoreceptors, Kif17 is necessary for outer segment (OS) development and disc morphogenesis, and this role has been recognized in mutant zebrafish and mice models [4,5]. However, little is known about the involvement of *KIF17* in human ophthalmological phenotypes.

The MAC spectrum (microphthalmia, anophthalmia, and coloboma) is a group of eye anomalies that comprises three different clinical entities:Microphthalmia refers to an eye with an axial length two standard deviations below the mean for age, and this condition has an estimated prevalence of 1 per 7000 live births.Anophtalmia is identified with the absence of ocular tissue in the orbit and has an estimated prevalence of 1 per 30,000 live births.Coloboma consists of an optic fissure closure defect and has an estimated prevalence of 1 per 5000 live births [6,7].

These anomalies may be unilateral or bilateral and can occur in any combination [8]. Some genes are associated with MAC, some with a confirmed pathogenicity, such as *SOX2*, *OTX2*, *PAX6*, *STRA6*, *ALDH1A3*, *RARB*, *VSX2*, *RAX*, and *FOXE3*; and others with a possible or uncertain pathogenicity, such as *BMP4*, *BMP7*, *GDF3*, *GDF6*, *ABCB6*, *ATOH7*, *C12orf57*, *TENM3*, and *VAX1* (Table 1). For that reason, molecular genetic testing can be requested to identify a genetic cause in patients with an ocular malformation in the MAC spectrum [7,8]. In the MAC spectrum, chorio retinal coloboma (CRC) is a rare congenital anomaly of part of the retinal pigment epithelium and choroid, due to incomplete closure of the embryonic fissure during fetal life [9]. CRC may either be bilateral or unilateral and is often associated with other ocular pathologies, particularly microphthalmia [9,10].

We report a Sicilian family with two siblings affected with an array of congenital eye anomalies consistent with the MAC spectrum in which clinical exome sequencing revealed biallelic segregating compound heterozygous variants in the *KIF17* gene associated with coloboma and microphthalmia.

## 2. Methods

After institutional review board approval of this study and informed consent from the family, we collected blood samples from the patients and their parents and extracted DNA using standard procedures. To investigate the genetic cause of the disease, a multi-gene panel containing 30 genes known to be implicated or possibly relevant to MAC spectrum and ocular anomalies were investigated through a multi-gene custom panel. DNA samples from the two affected siblings were enriched with Twist Custom Panel (clinical exome—Twist Bioscience) by NovaSeq6000 Illumina Platform. Raw data were analyzed using DRAGEN Germline Pipeline 3.3.7, and reads were mapped to the reference human genome sequence (GRch37/hg19). Single-nucleotide variants (SNVs) and short deletions or insertions (indels) were filtered according to genetic criteria for a very rare, highly penetrant autosomal dominant trait: (1) heterozygous; (2) MAF ≤ 0.0001, not reported in the GnomAD database; and (3) non-synonymous or affecting the splice site. Validation and parental origin of the variants were assessed by Sanger sequencing. To complete the genomic analysis of the family, whole exome sequencing (WES) was performed in both the affected sibling’s DNA (Figure 1A, II-1 and II-2). A Nextera rapid capture enrichment kit (Illumina) was used according to the manufacturer instructions. Libraries were sequenced on an Illumina HiSeq3000 using a 100-bp paired-end reads protocol. Sequence alignment to the human reference genome (UCSC hg19), and variant calling and annotation was performed, as described elsewhere [11,12]. The GnomAD database (https://gnomad.broadinstitute.org, accessed on 21 April 2021) was screened to assess the frequency of the identified variants (Table 2) in the general and European populations. We also screened a replication cohort of approximately 2000 individuals with motor and/or developmental delay sequenced by whole or clinical exome at “Giannina Gaslini” Children’s Hospital in Genoa, Italy.

## 3. Family Report

### 3.1. Phenotypic Features

Our patients are two siblings, both born from non-consanguineous and healthy parents (Figure 1A), and referred to the Unit of Genetics and Pediatrics, University Hospital “G. Martino” (Messina, Italy) for clinical evaluation due to congenital microphthalmia. The family history was unremarkable. The first sibling (II-1; Figure 1A) is currently an 8 year old boy, born full-term via cesarean section with a birth weight of 3420 g. As previously stated, during the very first days of life, left microphthalmia was noticed (Figure 1C,C′), which led to request a pediatric examination. At the neurological examination, fine motor as well as cognitive development were normal. He had a mild delay in acquiring sitting position and walking autonomously. No facial dysmorphisms were observed, and laboratory tests as well as electrocardiogram and hearing assessment were normal. Brain MRI showed bilateral coloboma with a cyst of the left eye (Figure 1C–C″). At the age of three years, the patient underwent an ophthalmological check-up, and fundus oculi examination confirmed bilateral coloboma of the optic nerve and a CRC of the left eye. His brother (II-2; Figure 1A) is a 1-year-old boy born full-term via cesarean section and with a birth weight of 2340 g. The neonatal period was normal except for an infection requiring a short cycle of antibiotics (ampicillin). However, also in this case, left microphthalmia was noticed (Figure 1D,D′). At the neurological examination, fine motor and cognitive development were normal. He had some mild delay in gross motor development, in reaching autonomous sitting position and walking. He had no facial dysmorphisms, and his laboratory tests were normal, as were electrocardiogram and hearing assessments. The right fundus oculi examination showed a posterior coloboma comprising the optic nerve, the choroid, and the retina; conversely, left fundus oculi examination could not be done due to highly severe microphthalmia. The ocular echography showed bilateral microphthalmia (L > R) and reduced size of the optic nerves (L > R). Brain MRI confirmed left microphthalmia with homolateral lens luxation, and bilateral coloboma without cysts (Figure 1D–D‴). The last ophthalmological evaluation was performed at age six months, and nystagmus was noticed while confirming left microphthalmia with homolateral lens luxation, left iris atrophy, right optic nerve, and CRC. In the same period, an ocular prosthesis was inserted in his left eye.

### 3.2. Genetic Results

Panel multi-gene sequencing revealed in both siblings a compound heterozygous variant in the *KIF17* gene [NM_020816.4: c.1255C > T (p.Arg419Trp); c.2554C > T (p.Arg852Cys)] that was confirmed to be a disease-candidate variant on clinical exome sequencing. In the GnomAD database (last accessed 21 April 2021), which contains exomes from 125,748 unrelated individuals, the p.Arg419Trp variant has an allele frequency of 0.0000607 in the general population (gnomAD), with a European allele frequency of 0.0000723 (Table 2). The p.Arg852Cys variant is not present in the gnomAD dataset. According to in silico tools (CADD, SIFT, PolyPhen, and GERP++), these variants are predicted to be likely deleterious. Other genes included in the panel (*PAX6*, *SOX2*, *GDF6*, *SHH*, *RAX*, *OTX2*, *VSX2*, *STRA6*, *RARB*, *CRYBA4*, *SIX6*, *SIX3*, *MFRP*, *GDF3*, *BMP4*, *PRSS56*, *SALL2*, *ALDH1A3*, *ABCB6*, *TENM3*, *BMP7*, *MAB21L2*, *VAX1*, *HCCS*, *BCOR*, *HMGB3*, *NAA10*, *RBP4*) were negative for candidate variants. WES data generated a total of 88,422,644 (II-1) and 86,235,564 (II-2) unique reads from the sequencing of the probands. Only indels and non-synonymous exonic/splicing variants shared by the two probands were kept and further filtered. In accordance with the pedigree and phenotype, priority was given to rare variants (<1% in public databases, including the 1000 Genomes project, NHLBI Exome Variant Server, Complete Genomics 69, and Exome Aggregation Consortium (ExAC v0.2)) that were fitting a recessive model (i.e., homozygous or compound heterozygous), and/or located in genes previously associated with ophthalmological phenotypes or developmental delay. After applying the above filtering criteria, no plausible homozygous variants were identified within the WES data. The compound heterozygous variant in *KIF17*, which was previously identified through multi-gene panel sequencing, emerged as the most likely explanation for the disease pathophysiology, as supported by gene expression and function in the eye (Figure 2A), conservation of the affected residues (Figure 2B), in silico predictors, co-segregation with the ocular phenotype of the probands, and the importance of this gene in eye development as supported by several animal studies. No other biallelic candidate variants in *KIF17* were identified in our replication cohort screening.

## 4. Discussion and Conclusions

Kif17 belongs to the kinesin-2 family of motor proteins, which is part of the kinesin superfamily of proteins (KIFs) and holds a variety of biological functions, including the regulation of the early development of the photoreceptor cilium [3,4,5,13]. As a motor protein, Kif17 shows an N-terminus head motor (N-kinesin) and a tail domain, and two putative stalk domains that form an alpha-helical coiled-coil (Figure 1B). The motor head contains a catalytic site for ATP hydrolysis and a binding site for microtubules. The alpha-helical coiled-coil stalk domain allows for protein–protein interactions and is connected to the head domain via a family-specific flexible neck linker. The tail domain binds cargo, but in the absence of cargo, Kif17 undergoes auto-inhibition by folding about its central hinge to suppress motility. The folding enables the coiled-coil 2 (CC2) segment and the C-terminal tail domain to interact directly with the dimeric head motor domain [3].

Our variants fall within the CC1 and tail domains and hence may alter these auto-inhibition and folding processes. Moreover, the Kif17 protein is also involved in the transport of vesicles containing the N-methyl-D-aspartate receptor NR2B subunit from the cell body to dendrites, a process specifically associated with glutamatergic neurotransmission, learning, and memory [14,15]. Interestingly, animal models of deficits in learning and enhanced working and spatial memory due to reduced and increased expression of *KIF17*, respectively, are reported in the literature [16,17]. Thus, our findings highlight *KIF17* as a gene potentially involved also in both ocular morphogenesis and neurodevelopmental processes. Of note, an association between *de novo* loss-of-function variants in *KIF17* and neurodevelopmental phenotypes, including abnormal behavior and schizophrenia, was previously proposed [18]. Different genes encoding kinesins and motor proteins have been previously implicated in brain development, and defining the full spectrum of disease-causing molecular pathways will help to diagnose, monitor, and accelerate treatment development in genetic neurodevelopmental conditions with associated cytoskeleton alterations or modulating ionotropic glutamate receptor subunits [19,20,21,22,23,24]. 

In this work, we describe two siblings with mild motor developmental delay (but no behavioral abnormalities) who displayed different ophthalmologic anomalies and were found to carry compound heterozygous variants in the *KIF17* gene.

The non-synonymous *KIF17* variants identified in this study were the most plausible candidate genetic variants in our family, based on their segregation with the phenotype of the affected siblings and the predicted deleterious consequences, as supported by several in silico tools. Functional (cellular) studies using patient-derived fibroblasts to assess the impact of the identified missense variants on transcript and protein were not possible, due to the absence of *KIF17* expression in skin. However, different animal model studies, including from mice and zebrafish, have highlighted an essential role of *KIF17* in eye development, particularly for the OS development and disc morphogenesis [3,4,5].

In vertebrates, eye formation is the result of coordinated induction and differentiation processes during embryogenesis. Disruption of any one of these events has the potential to cause ocular growth and structural defects, such as the MAC spectrum anomalies. Their etiology includes genetic and environmental factors; several hundred genes involved in ocular development have been identified in humans or animal models, although many of these genes have been described in single cases or families, and some genetic syndromes include MAC anomalies occasionally as part of a wider phenotype [18]. As part of the complex eye embryogenesis, the normal development of OS photoreceptors is also essential for visual function in vertebrates. Defects in Kif17 seem to have been associated with developmental abnormalities of the photoreceptor OS, a highly modified and specialized primary cilium. Of note, different animal models with defects in other genes primarily involved in the development of the OS photoreceptor have been described with microphthalmia and additional anterior/posterior segment anomalies; importantly, differences may occur in the phenotypic presentation and the extension of the ocular structural defects across different disease models and species [25,26]. Interestingly, many defects in the structure and function of the photoreceptor primary cilium are associated with a group of human inherited conditions known as "ciliopathies", and MAC anomalies can be sometimes observed as part of the ocular phenotype of these patients [27]. In this context, it is possible that *KIF17* plays a role beyond the organization and turnover of OS photoreceptors, and we may speculate its potential involvement in the development of different eye segments during the eye development of vertebrates. Similar to other cilium-related genes with a predominant OS expression and function, some inter-species differences may occur also in *KIF17*-related ocular phenotypes, explaining the complex presentation that we observed in our family. Thus, based on previous animal studies and the present findings, we suggest a potential association of the p.Arg419Trp and p.Arg852Cys variants with the congenital coloboma and microphthalmia observed in our affected siblings. However, the interpretation of the role of these compound heterozygous variants remains challenging due to the lack of previously described MAC cases carrying biallelic variants in this gene. In addition, some other genetic factors and/or mechanisms causing (or contributing to) MAC phenotype in our family may have not been identified (or covered) due to technical limitations of the genetic analysis we performed (focusing on nucleotide variants within the coding regions of clinically relevant genes). Thus, further clinical and experimental studies will be required to confirm the implication of *KIF17* in human ocular phenotypes. In this regard, the integration of *KIF17* in multi-gene panels for the investigation of individuals affected with coloboma and microphthalmia, as well as the *KIF17* screening within existing and future exonic and genomic datasets generated from undiagnosed MAC individuals, may help to confirm the possible involvement of *KIF17* in autosomal recessive ocular anomalies.

## Figures and Tables

**Figure 1 ijms-22-04471-f001:**
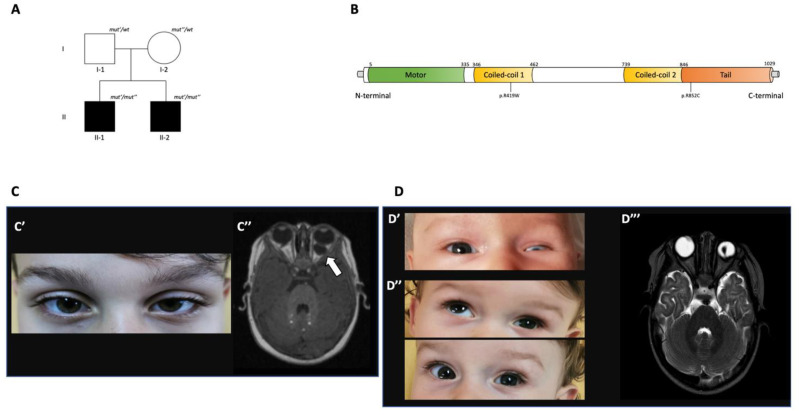
Family tree, KIF17 protein structure, and the clinico-radiological features of our siblings. (**A**) Pedigree from family. (**B**) KIF17 protein structure with domains and the two pathogenic variants found in our probands. (**C**) Clinico-radiological findings of Patient II-1. (**C′)** Image of eyes showing left microphthalmia at age. (**C″**) Brain MRI at age 7 months. Axial FLAIR images demonstrate bilateral focal posterior defects of the globes with vitreous herniation suggestive for bilateral posterior coloboma. In the left eye a fluid-density cist is also present (white arrow). (**D**) Clinico-radiological features of Patient II-2. (**D′**) Image of eyes showing left microphthalmia at age. (**D″**) Additional eye images taken at age. (**D‴**) Brain MRI at age 5 months. Axial T2-weighted image showing right “egg shaped” globe suggestive for posterior coloboma. In the left eye the “egg shape” is less deducible due to microphthalmia, as well as lens luxation.

**Figure 2 ijms-22-04471-f002:**
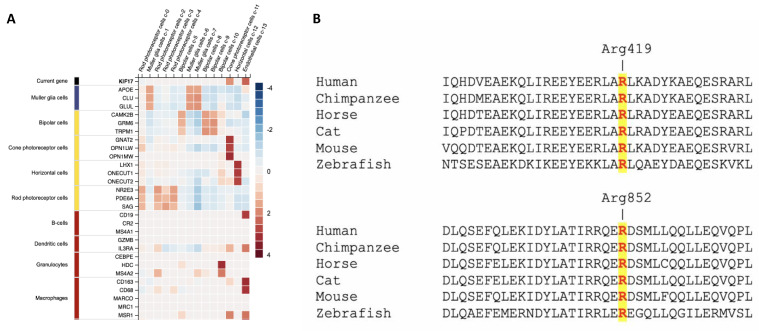
(**A**) *KIF17* gene expression in the different eye-related cell types based on the Human Protein Atlas (www.proteinatlas.org, accessed on 21 April 2021). (**B**) Multiple alignment showing complete conservation across species and KIF17 homolog of the residues affected by the variants identified in this study (these variants are highlighted in yellow). Human KIF17 (UniProt: Q9P2E2), Chimpanzee KIF17 (UniProt: H2PY89), Horse KIF17 (UniProt: F6VCL8), Cat KIF17 (UniProt: A0A2I2V0D2), Mouse KIF17 (UniProt: Q99PW8), Zebrafish KIF17 (UniProt: F1QV34).

**Table 1 ijms-22-04471-t001:** Pathogenic variants frequently associated with MAC spectrum [8].

Gene	Percentage of MAC Individuals with Pathogenic Variants in This Gene
*SOX2*	15–20%
*OTX2*	2–5%
*RAX*	3%
*FOXE*	3%
*BMP4*	2%
*PAX6*	2%
*BCOR*	>1%
*CHD7*	>1%
*STRA6*	>1%
*GDF6*	1%

**Table 2 ijms-22-04471-t002:** Frequency information of the *KIF17* p.Arg419Trp variant based on the GnomAD database.

Population	Allele Count	Allele Number	Number of Homozygotes	Allele Frequency
East Asian	2	19,836	0	0.0001008
Latino/American	3	35,348	0	0.00008487
South Asian	2	30,584	0	0.00006539
European (non-Finnish)	8	125,508	0	0.00006374
European (Finnish)	1	24,992	0	0.00004001
African/African-American	0	23,632	0	0.000
Ashkenazi Jewish	0	10,270	0	0.000
Other	0	7108	0	0.000
XX	9	125,720	0	0.00007159
XY	7	151,558	0	0.00004619
Total	16	277,278	0	0.00005770

## Data Availability

The data presented in this study are available on request from the corresponding author.

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
