# Peer review of "Biallelic Variants in KIF17 Associated with Microphthalmia and Coloboma Spectrum"

_ijms, 2021, doi:10.3390/ijms22094471_

Round 1
Reviewer 1 Report
The work by Riva A et al. describe biallelic missense KIF17 variants associated with a coloboma microphthalmia spectrum.
Scientific knowledge and previous animal studies identified the crucial role of KIF in the developing eye in different models. The merit of the present manuscript is to highlight the potential involvement of KIF17 also in human ocular phenotypes. The two low-frequency missense variants identified by the authors affect conserved residues of the Kif17 protein and are predicted deleterious by several in silico tools. The variant segregate with the disase within the family. The phenotype described by the authors is convincing and some mild motor developmental delay was also present in association with coloboma and microphthalmia, supporting a role of Kinesin 17 also in brain development.
The manuscript is well written overall. The introduction is succinct and informative and the discussion and conclusions are compelling. I have some minor suggestions to improve the paper especially in the methods section.
I understand a detailed mechanicistic functional characterization is beyond the scope of the present family report and that is also limited by the difficulties to assess impact of the mutations on a cellular level (due to the restricted expression patterns of the gene). However, the bioinformatic analysis should be slightly deepened e.g., including a table with frequency informations of these variants in both European and possibly Arabic populations (given that family is originally from Sicily island with a frequent Arabic background). Also, despite the variants are well conserved in evolution, an allignment of the residues inter-species would be important and could be integrated in the present figure 1.
Finally, the present reviewer understand the difficulties of identifying multiple families carrying novel or rare genetic defects. However, the authors could include in their methods section any screening of existing genomic datasets they attempted with the aim of identifying further cases carrying biallelic KIF17 variants. I understand the effect of the mutations can be residue-specific but I think a more detailed replication cohort screening would be beneficial to this work and would add further genetic consistency.
Author Response
Point 1: The work by Riva A et al. describe biallelic missense KIF17 variants associated with a coloboma microphthalmia spectrum. Scientific knowledge and previous animal studies identified the crucial role of KIF in the developing eye in different models. The merit of the present manuscript is to highlight the potential involvement of KIF17 also in human ocular phenotypes. The two low-frequency missense variants identified by the authors affect conserved residues of the Kif17 protein and are predicted deleterious by several in silico tools. The variant segregate with the disase within the family. The phenotype described by the authors is convincing and some mild motor developmental delay was also present in association with coloboma and microphthalmia, supporting a role of Kinesin 17 also in brain development. The manuscript is well written overall. The introduction is succinct and informative and the discussion and conclusions are compelling. I have some minor suggestions to improve the paper especially in the methods section.
Response 1: “We are grateful to the Reviewer for the very positive comments related to our manuscript.”
Point 2: I understand a detailed mechanicistic functional characterization is beyond the scope of the present family report and that is also limited by the difficulties to assess impact of the mutations on a cellular level (due to the restricted expression patterns of the gene). However, the bioinformatic analysis should be slightly deepened e.g., including a table with frequency informations of these variants in both European and possibly Arabic populations (given that family is originally from Sicily island with a frequent Arabic background).
Response 2: “Thank you very much for your positive comments. Indeed a detailed mechanicistic characterization is beyond the scope of the present work. We implemented now the genetic methodology of our study according to your comments. The paragraph was modified as suggested and a table with alleles frequency in European and African populations is now included (Table 2).”
Point 3: Also, despite the variants are well conserved in evolution, an allignment of the residues inter-species would be important and could be integrated in the present figure 1.
Response 3: “Thank you for this suggestion. We integrated now a novel figure (Figure 2) showing some expression data of KIF17 in different cell types within the eye and also an inter-species alignments of KIF17 gene comparing different species. This confirm complete conservation of the residues affected by the mutations we identified in our patients.”
Point 4: Finally, the present reviewer understand the difficulties of identifying multiple families carrying novel or rare genetic defects. However, the authors could include in their methods section any screening of existing genomic datasets they attempted with the aim of identifying further cases carrying biallelic KIF17 variants. I understand the effect of the mutations can be residue-specific but I think a more detailed replication cohort screening would be beneficial to this work and would add further genetic consistency.
Response 4: “Thank you very much for this important point. In addition to more detailed information on the GnomAD database screening we also screened genomic (exonic) dataset from a large cohort of children with neurodevelopmental impairment recruited and sequenced at Children’s Hospital “G. Gaslini” and did not identify other biallelic variant in KIF17.”

Reviewer 2 Report
The study by Riva and colleagues presents two cases of congenital coloboma, microphthalmia and mild motor developmental delay associated to defective KIF17. This is a novel association, relevant to expand our understanding of the molecular defects underlying MAC conditions, and very interesting due to the nature of this gene and the many functions during development it could be involved in. The study is well written and the findings clearly reported. My only comment regard the discussion of KIF17 function:
KIF17 is a kinesin, a MT motor protein, and as such it is likely to have many roles during embryonic development and in adult tissues. In the eye, the authors highlight the previously described role of this protein for photoreceptors outer segment development. Intriguingly, not much more is known about the role of this kinesin in the eye, but it seems to this reviewer difficult to relate defects in photoreceptor differentiation with MAC phenotypes. Indeed, MAC phenotypes are more likely related to defective morphogenesis of the eye primordium prior to the stages at which photoreceptors differentiate. The manuscript would benefit enormously from a more extensive discussion of potential roles for this gene. Is there information regarding KIF17 expression during embryonic development in model organisms? Is there the potential for this protein to have a role during earlier stages of eye formation, or in other cell types apart from photoreceptors? Are there any other roles described in other organs, which could be mentioned and could provide clues as to potential roles of this protein during eye morphogenesis?
Incorporating a more extensive discussion will allow the reader to put in an appropriate context the potential function of KIF17 during eye formation.
Author Response
Point 1: The study by Riva and colleagues presents two cases of congenital coloboma, microphthalmia and mild motor developmental delay associated to defective KIF17. This is a novel association, relevant to expand our understanding of the molecular defects underlying MAC conditions, and very interesting due to the nature of this gene and the many functions during development it could be involved in. The study is well written and the findings clearly reported. My only comment regard the discussion of KIF17 function.
Response 1: “We are grateful to the Reviewer for the very positive comments related to our manuscript”
Point 2: KIF17 is a kinesin, a MT motor protein, and as such it is likely to have many roles during embryonic development and in adult tissues. In the eye, the authors highlight the previously described role of this protein for photoreceptors outer segment development. Intriguingly, not much more is known about the role of this kinesin in the eye, but it seems to this reviewer difficult to relate defects in photoreceptor differentiation with MAC phenotypes. Indeed, MAC phenotypes are more likely related to defective morphogenesis of the eye primordium prior to the stages at which photoreceptors differentiate. The manuscript would benefit enormously from a more extensive discussion of potential roles for this gene. Is there information regarding KIF17 expression during embryonic development in model organisms? Is there the potential for this protein to have a role during earlier stages of eye formation, or in other cell types apart from photoreceptors? Are there any other roles described in other organs, which could be mentioned and could provide clues as to potential roles of this protein during eye morphogenesis?Incorporating a more extensive discussion will allow the reader to put in an appropriate context the potential function of KIF17 during eye formation.
Response 2: “Thank you very much for these important considerations. We now did an effort to review model organisms through the literature in light of further dissecting the potential function of KIF17 during early eye development. We implemented the discussion section discussing the different genes implicated in the trafficking or the function at the photoreceptor outer segment cilium and implicated in a range of ocular anomalies ranging between OS disorganization to coloboma and anophthalmia/microphthalmia. We included now several considerations on the potential role of Kif17 in the development of different eye segments and discussed the potential different phenotypic expression within the same organisms as well as across different species. This is similar to other genes/phenotypes associated with MAC anomalies often in the context of defects in cilium-related genes. Thus, we highlighted the possible role of KIF17 in human brain and function and included some comments on embryogenesis and timelines of eye development.”
